# COVID-19 mitigation measures in primary schools and association with infection and school staff wellbeing: An observational survey linked with routine data in Wales, UK

Emily Marchant[1,2]*, Lucy Griffiths[1], Tom Crick[3], Richard Fry[1,2], Joe Hollinghurst[1], Michaela James[1,2], Laura Cowley[2,4], Hoda Abbasizanjani[1], Fatemeh Torabi[1], Daniel A. Thompson[1], Jonathan Kennedy[1,2], Ashley Akbari[1], Michael B. Gravenor[1], Ronan A. Lyons[1], Sinead Brophy[1,2]

1 Population Data Science, Medical School, Swansea University, Swansea, United Kingdom, 2 National Centre for Population Health and Wellbeing Research, Swansea University, Swansea, United Kingdom, 3 School of Education, Swansea University, Swansea, United Kingdom, 4 Research and Evaluation Division, Public Health Knowledge and Research Directorate, Public Health Wales, Cardiff, United Kingdom

* E.K.Marchant@swansea.ac.uk

**Data Availability Statement:** The routine data used in this study are available from the SAIL Databank at Swansea University, Swansea, UK. All proposals

## Abstract

### Introduction

School-based COVID-19 mitigation strategies have greatly impacted the primary school day (children aged 3–11) including: wearing face coverings, two metre distancing, no mixing of children, and no breakfast clubs or extra-curricular activities. This study examines these mitigation measures and association with COVID-19 infection, respiratory infection, and school staff wellbeing between October to December 2020 in Wales, UK.

### Methods

A school staff survey captured self-reported COVID-19 mitigation measures in the school, participant anxiety and depression, and open-text responses regarding experiences of teaching and implementing measures. These survey responses were linked to national-scale COVID-19 test results data to examine association of measures in the school and the likelihood of a positive (staff or pupil) COVID-19 case in the school (clustered by school, adjusted for school size and free school meals using logistic regression). Linkage was conducted through the SAIL (*Secure Anonymised Information Linkage*) Databank.

### Results

Responses were obtained from 353 participants from 59 primary schools within 15 of 22 local authorities. Having more direct non-household contacts was associated with a higher likelihood of COVID-19 positive case in the school (1–5 contacts compared to none, OR 2.89 (1.01, 8.31)) and a trend to more self-reported cold symptoms. Staff face covering was not associated with a lower odds of school COVID-19 cases (mask vs. no covering OR 2.82 (1.11, 7.14)) and was associated with higher self-reported cold symptoms. School staff reported the impacts of wearing face coverings on teaching, including having to stand closer

to use SAIL data are subject to review by an IGRP. Before any data can be accessed, approval must be given by the IGRP. The IGRP gives careful consideration to each project to ensure proper and appropriate use of SAIL data. When access has been approved, it is gained through a privacy-protecting safe haven and remote access system referred to as the SAIL Gateway. SAIL has established an application process to be followed by anyone who would like to access data via SAIL at: https://www.saildatabank.com/application-process. This study has been approved by the IGRP as project 0911.

**Funding:** The Economic and Social Research Council (ESRC) funded the development of the HAPPEN network (grant number: ES/J500197/1) which this research was conducted through (https://esrc.ukri.org/). The National Centre for Population Health and Wellbeing Research (NCPHWR) funded by Health and Care Research Wales (https://healthandcareresearchwales.org/) provided infrastructural support for this work. This work was supported by the Con-COV team funded by the Medical Research Council (grant number: MR/V028367/1) (https://www.ukri.org/councils/mrc/). This work was supported by Health Data Research UK, which receives its funding from HDR UK Ltd (HDR-9006) funded by the UK Medical Research Council, Engineering and Physical Sciences Research Council, Economic and Social Research Council, Department of Health and Social Care (England), Chief Scientist Office of the Scottish Government Health and Social Care Directorates, Health and Social Care Research and Development Division (Welsh Government), Public Health Agency (Northern Ireland), British Heart Foundation (BHF) and the Wellcome Trust. This work was a collaboration with the ADR Wales programme of work. ADR Wales is part of the Economic and Social Research Council (part of UK Research and Innovation) funded ADR UK (grant ES/S007393/1). This work was supported by the Wales COVID-19 Evidence Centre, funded by Health and Care Research Wales. The funders had no role in study design, data collection and analysis, decision to publish, or preparation of the manuscript.

**Competing interests:** The authors declare there are no competing interests.

**Abbreviations:** SAIL Databank, Secure Anonymised Information Linkage; SARS-CoV-2, Severe acute respiratory syndrome coronavirus 2; PCR, Polymerase chain reaction; HAPPEN, Health and Attainment of Pupils in a Primary Education Network; SWAC, School Workforce Annual Census; PLASC, Pupil Level Annual School Census;

to pupils and raise their voices to be heard. 67.1% were not able to implement two metre social distancing from pupils. We did not find evidence that maintaining a two metre distance was associated with lower rates of COVID-19 in the school.

## Conclusions

Implementing, adhering to and evaluating COVID-19 mitigation guidelines is challenging in primary school settings. Our findings suggest that reducing non-household direct contacts lowers infection rates. There was no evidence that face coverings, two metre social distancing or stopping children mixing was associated with lower odds of COVID-19 or cold infection rates in the school. Primary school staff found teaching challenging during COVID-19 restrictions, especially for younger learners and those with additional learning needs.

## Introduction

The COVID-19 global pandemic caused by the transmission of severe acute respiratory syndrome coronavirus 2 (SARS-CoV-2) resulted in the temporary closure of educational settings worldwide [1]. Implemented worldwide from mid-April 2020, school closures were used as a public health measure to reduce social contacts and the risk of transmission amongst pupils, school staff, families and the wider community. However, recent evidence indicates that children below the age of 14 appear to have lower susceptibility to infection and display fewer clinical symptoms [2–5]. Population-level data suggests that whilst transmission risks within school exists, risks are lower compared to within households [6]. Adults living with young children (0–11 years) during the period after schools reopened encountered no greater risk of COVID-19 infection [7], and school staff were at no greater risk of COVID-19 infection than other working-age adults [8].

Educational settings reopened for face-to-face teaching and learning from September to December 2020. In Wales, one of the four nations of the UK, education is a devolved responsibility of the Welsh Government. Operational guidance to schools in Wales in the 2020 autumn term [9] (1 September to 22 December) included widespread adaptation to social behaviours and a variety of school-based mitigation measures. This included encouraging wearing face coverings, reducing contacts, maintaining social distancing between pupils and staff, segregating classes and guidance on breakfast clubs, extra-curricular activities and outdoor learning [9].

Research examining the implementation of guidelines by schools highlights major challenges, including the ability of school staff to maintain a two metre distance from staff and pupils [10]. School staff highlight the conflict between balancing preventative measures with learning, with measures such as physical distancing policies negatively impacting on teaching quality. A rapid scoping review assessing the impacts of school-based measures concluded that there is an urgent need for research assessing the effectiveness of these measures on directly affected populations (e.g. pupils and school staff) [11], and on the psychosocial well-being and mental health of school populations. This is important as evidence suggests teacher wellbeing is a critical factor in creating stable environments for children to thrive [12] and is positively associated with academic achievement [13].

This study linked routinely collected COVID-19 polymerase chain reaction (PCR) test results data with survey data to examine the association between COVID-19 positive cases within the primary school setting and different school-based mitigation measures aligned to

IGRP, Information Governance Review Panel;
GAD-7, Generalized Anxiety Disorder; PHQ-9,
Patient Health Questionnaire; ALF, Anonymous
Linking Field; OR, Odds Ratio; ALN, Additional
learning needs; EAL, English as an additional
language; WHO, World Health Organization.

guidance, implemented between October to December 2020. It also examined these measures with school staff's self-reported (a) cold symptoms in the previous seven days, as a proxy for infection rates; and (b) levels of anxiety and depression. Secondary qualitative data exploring the impacts of wearing face coverings are also presented to complement quantitative findings.

## Methods

### Study design

This study adopted a mixed methods design. Participants were recruited through the HAPPEN primary school network (*Health and Attainment of Pupils in a Primary Education Network*) [14] in September 2020. An online survey (open 9 October 2020 to 16 December 2020) with school staff captured self-reported implementation of school-based COVID-19 mitigation measures and individual level outcomes of cold symptoms and anxiety/depressive symptoms. The survey findings were linked with routine data on COVID-19 test results for staff and pupils within the respective school of the staff participant for the school-level outcome. Linkage was performed using the SAIL (*Secure Anonymised Information Linkage*) Databank [15,16]. Data were linked at the individual level using the School Workforce Annual Census (SWAC) to assign each school staff to their school, and the Pupil Level Annual School Census (PLASC) to identify pupil by school and link COVID-19 test results to the appropriate school [17]. In addition, open-ended survey responses were used to examine views of school staff using a content analysis approach [18,19]. The RECORD checklist [20] for this study is presented in S1 Appendix.

### Ethics

Ethical approval was granted by the Swansea University Medical School Research Ethics Committee (2017-0033E). Information sheets and consent forms were distributed via email to participants detailing the aims of the study. To participate in the survey, primary school staff were required to provide written informed consent. All participants were able to withdraw from the research at any point. All participants were assigned a unique ID number, and any personal data such as names were removed. Electronic data (survey responses) were stored in password-protected files that were only accessible to the research team. The routine data used in this study are available in the SAIL Databank [21] and are subject to review by an independent Information Governance Review Panel (IGRP), to ensure proper and appropriate use of SAIL data. Before any data can be accessed, approval must be received from the IGRP. When access has been approved, it is accessed through a privacy-protecting safe haven and remote access system referred to as the SAIL Gateway. SAIL has established an application process to be followed by anyone who would like to access data via SAIL. This study has been approved by the SAIL IGRP (project reference: 0911).

### School staff survey and linked data

A convenience sample of primary school staff were recruited by contacting members of the HAPPEN network and directly emailing all primary schools in Wales, UK (n = 1,203) in September 2020. The survey was promoted through existing partnerships with stakeholders including regional education consortia groups. The online survey was open for responses from 9 October 2020 to 16 December 2020 (study period) when schools returned for face-to-face teaching. Inclusion criteria for participation was any primary school staff working within a local authority maintained (publicly funded) primary school. The development of the survey was based on input from the research team specialising in child health and education research

(authors EM, MJ, SB), education stakeholders (regional education consortia curriculum staff) and a headteacher and teacher from two primary schools to ensure appropriate wording and usability. The final survey contained 41 questions consisting of demographic, categorical and open-ended questions. The survey included the validated questionnaires Generalized Anxiety Disorder (GAD-7) [22] and Patient Health Questionnaire (PHQ-9) [23] to assess the presence and severity of anxiety and depressive symptoms. The survey was conducted online and could be completed by a member of school staff at a convenient time via an electronic device including mobile phone, tablet, laptop and computer. Responses were downloaded to an Excel spreadsheet. Quantitative data responses were uploaded to the SAIL Databank [15,16] to be linked with COVID-19 school testing data [17], and analysed using Stata (version 16). A full copy of the survey is presented in S2 Appendix, and detail regarding survey item, item response categories and item coding for analyses are presented in S3 Appendix.

The process of data coding involved two researchers. The first researcher downloaded the raw data, cleaned the data, checked for duplicates, generated a unique participant ID number and removed identifiable information. This process protects participants' anonymity by ensuring that the second researcher conducting the analyses could not identify individuals. This coded dataset was uploaded to the SAIL Databank, a national data infrastructure asset of anonymised data about the population of Wales that enables secure data linkage and analysis for research. To link the data, the demographic data are separated from the survey data and sent to a trusted third party, Digital Health and Care Wales and the survey data goes to SAIL using a secure file upload. A unique Anonymous Linking Field (ALF) is assigned to the person-based record before it is joined to clinical data via a system linking field. This dataset was accessible to authors listed from Population Data Science.

## Quantitative analysis

A COVID-19 school incident in Wales, UK, is defined as one or more positive COVID-19 cases in a school [24]. The primary outcome was the probability of at least one positive school-level COVID-19 test (pupils or staff) within the school setting linked to the staff participant during the study period. Secondary binary outcomes investigated at an individual level captured by the online survey included self-reported cold symptoms in the previous seven days as a proxy of infection risk as evidence suggests a crossover of symptoms between COVID-19 and the common cold [25], and moderate/severe anxiety (GAD-7) and moderate/severe depression (PHQ-9). Eligibility criteria within final analyses models were any primary school staff participant with complete linked survey and routine records. Participants contracted to multiple schools were excluded from analyses (n = 3) (see Fig 1).

Logistic regression analyses adjusting for confounding variables (school size, proportion of pupils eligible for free school meals as an indicator for deprivation) and clustered by school determined the Odds Ratio (OR) at a school level for at least one positive linked COVID-19 test at the respective school during the study period and for individual-level (school staff) secondary outcomes (self-reported cold symptoms, moderate/severe anxiety, moderate/severe depression).

All exposure measures relating to government guidance were captured through self-report by school staff via the online survey and were analysed in individual models (univariable) and then in a combined model (multivariable). Items with multiple category responses or continuous numerical values were assigned ordinal categories to ease interpretation. For example survey response categories for *keep two metres from pupils/staff* included i) *never*, ii) *rarely*, iii) *some of the time*, iv) *most of the time*, v) *always*, with combined ordinal categories for analyses of i) *never/rarely*, ii) *some of the time*, iii) *most of the time/always*. For these variables, likelihood-ratio tests of variables as whole were performed to assess goodness of fit between models

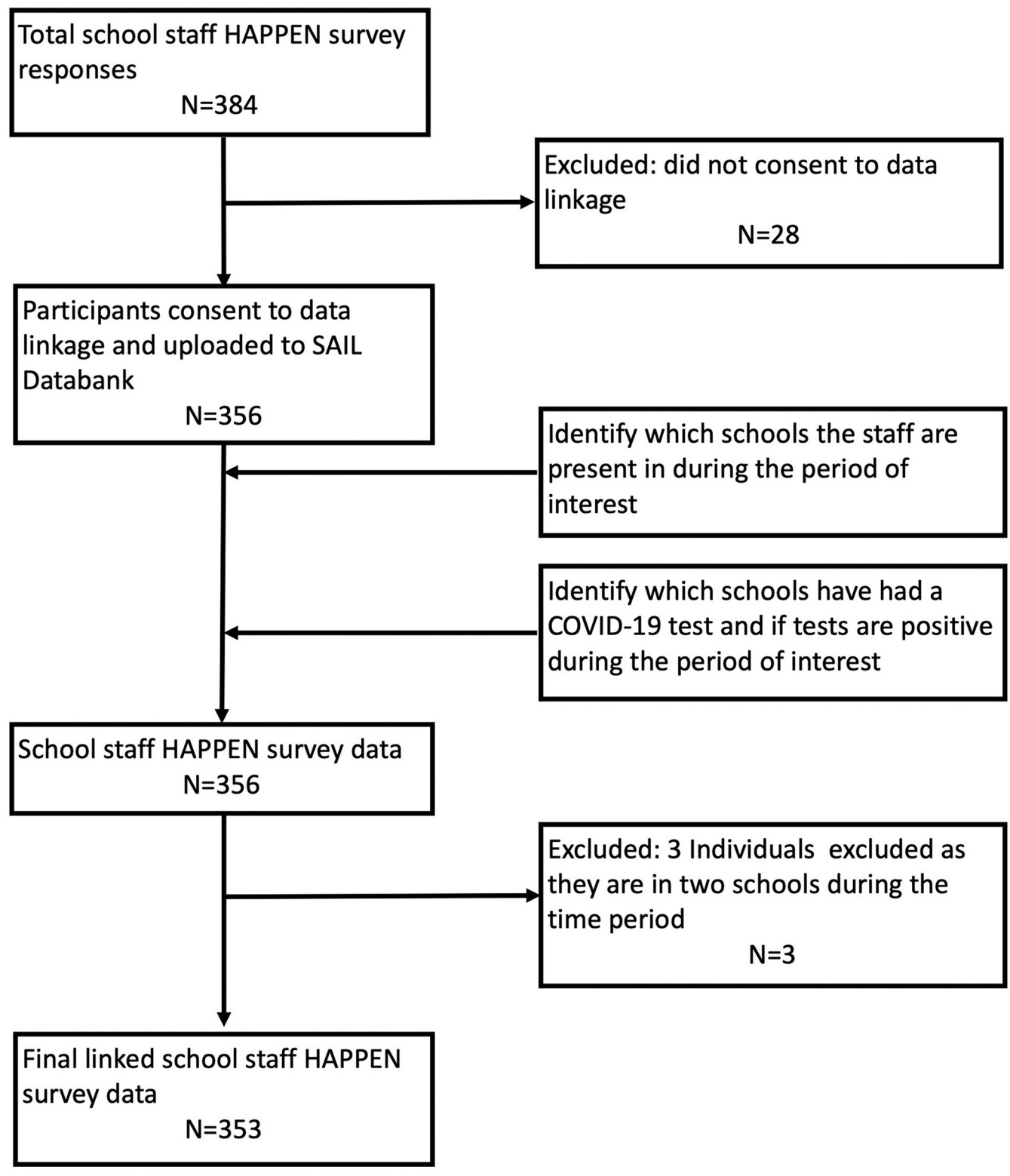

**Fig 1. Cohort flow diagram.**

including and excluding variables for the primary outcome. Further detail of exposures for all survey items within analyses including possible response category, grouping and coding can be found in S3 Appendix. This study assumed self-reported mitigation measures to be in effect for the duration of the study period based on operational guidance issued to schools at the time of the study [9].

### Qualitative analysis

Secondary qualitative content analysis was conducted to explore the impact of wearing face coverings on teaching, attained from item 29 (see S2 Appendix). Content analysis aims to make contextual inferences of data by condensing text into related concepts to provide knowledge to describe a phenomenon [18]. Conceptual content analysis was chosen to quantify the frequency of reoccurring words/themes and offer a descriptive lens of the quantitative data in terms of the most significant impacts of wearing face coverings for school staff [26,27]. An inductive approach was used as knowledge of this subject is limited due to the new and rapidly evolving nature of the COVID-19 pandemic. The lead researcher (EM) followed the steps of preparation, organising and reporting outlined by Elo and Kyngäs [26].

During the preparation stage, words or sentences were chosen as the unit of analysis to represent related concepts. The lead researcher (EM) who was female and had previous experience in qualitative data analysis read the open-ended responses several times to facilitate immersion in the data [28] and to gain an understanding of 'what is going on' [29]. The use of memoing recorded notes of patterns and emerging insights relating to coding ideas. Thoughts relating to decision processes were documented in a reflexive journal [30,31]. In the case of inductive content analysis, an open coding process to organising the data was applied by manually assigning freely generated open codes, consisting of words and sentences representing key conceptual responses. The initial list of words and sentences were grouped under higher order headings [28], with each heading named using content-characteristic words that describe the phenomenon [26]. The categories produced were discussed and reviewed with the research team to develop the final list of category headings characterising any impacts of face coverings on teaching. The researchers did not have any interaction with participants.

### Results

Reponses were obtained from 353 participants from 59 primary schools located within 15 local authorities in Wales, UK (Table 1). A cohort flow diagram is presented in Fig 1. 87 (24.7%) participants had a linked COVID-19 positive test, 31 (8.8%) reported cold symptoms, 62 (17.6%) and 67 (19.0%) reported moderate/severe anxiety and depression respectively. Participants were removed from the regression analyses due to missing values for the following outcomes; cold symptoms outcomes (n = 8), anxiety (n = 49) and depression (n = 125) (multivariable models). Missing values of exposure variables ranged from 0 to 19 (see Table 2). Complete case analyses are presented below. Sensitivity analyses where missing responses are coded as 0 are presented in S4 Appendix.

### Quantitative results

Exposure variables were examined individually (univariable) for association with outcomes and then all variables were entered together (multivariable) in the final combined models for the outcomes of school-level COVID-19 (Table 3), self-reported cold symptoms (Table 4), moderate/severe anxiety (Table 5) and depressive (Table 6) symptoms. Models were adjusted for school size and free school meal proportion, and clustered by school (see S3 Appendix for exposure response coding).

**Table 1. Demographics of survey respondents;** *obtained from Welsh Government data online [32].

| Characteristics | % (n) |
|---|---|
| **Number of participants (school staff)** | 353 |
| **Number of schools** | 59 *(1,203 national total*)* |
| **Number of local authorities** | 15 *(22 national total*)* |
| **School characteristics** | |
| **Mean Percentage of Free School Meals** | 20.6% *(national average 19%*)* |
| **Free School Meal category** | |
| *0–10%* | 28.8% (17) |
| *11–20%* | 25.4% (15) |
| *21–30%* | 23.7% (14) |
| *31%+* | 22.1% (13) |
| **School size (number of pupils)** | *(national average 223*)* |
| *0–100* | 8.5% (5) |
| *101–200* | 32.2% (19) |
| *201–300* | 23.7% (14) |
| *301–400* | 16.9% (10) |
| *401–500* | 15.3% (9) |
| *501+* | 3.4% (2) |
| **Participant characteristics** | |
| **Job role** | |
| *Support staff* | 4.1% (14) |
| *Supply teacher* | 1.2% (4) |
| *Teaching assistant* | 35.1% (120) |
| *Teacher* | 53.2% (182) |
| *Headteacher (teaching)* | 1.2% (4) |
| *Headteacher (non-teaching)* | 5.3% (18) |
| *Missing* | 3.2% (11) |
| *Full time* | 78.8% (278) |
| *Part time* | 18.4% (65) |
| *Missing* | 2.8% (10) |
| **Year group** | |
| *Foundation phase (ages 3–7) Reception* | 25.6% (90) |
| *Key Stage 2 (ages 7–11)* | 30.0% (106) |
| *Combination of years* | 35.7% (126) |
| *Missing* | 8.8% (31) |
| **Outcomes** | |
| **Positive COVID-19 school test** | 24.7% (87) |
| Missing | 0 |
| **Report cold symptoms previous 7 days** | 8.8% (31) |
| Missing | *2.3% (8)* |
| **Report moderate/severe anxiety (GAD-7)** | 17.6% (62) |
| Missing | *13.9% (49* |
| **Report moderate/severe depression (PHQ-9)** | 19.0% (67) |
| Missing | 35.4% (125) |

**Number of non-household contacts (1-metre, direct).** In the multivariable models, compared to reporting 0 contacts, reporting more non-household direct contacts was associated with higher odds of COVID-19 at the school level (1–5 contacts OR = 2.89, Table 3, model 1),

**Table 2. Distribution of individual school staff responses to mitigation survey items and school-level response agreement (see S3 Appendix).**

| Survey item | Response | % (n) | % (n) of schools with ≥80% agreement of responses (for school-level outcome) |
|---|---|---|---|
| **Keep two metres from pupils** | *Never/rarely* | 67.1% (237) | 61% (36) |
| | *Sometimes* | 23.5% (83) | |
| | *Most of the time/always* | 7.9% (28) | |
| | *Missing* | 1.4% (5) | |
| **Keep two metres from staff** | *Never/rarely* | 9.1% (32) | 59% (35) |
| | *Sometimes* | 22.1% (78) | |
| | *Most of the time/always* | 66.9% (236) | |
| | *Missing* | 2.0% (7) | |
| **Wear face covering** | *No* | 56.1% (198) | 83% (49) |
| | *Mask* | 31.4% (111) | |
| | *Visor* | 11.3% (40) | |
| | *Missing* | (<5) | |
| **Non-household contacts within one metre** | *0* | 24.7% (87) | 41% (24) |
| | *1–5* | 38.8% (137) | |
| | *≥ 6* | 36.5% (129) | |
| | *Missing* | 0 | |
| **Non-household contacts direct** | *0* | 81.9% (289) | 73% (43) |
| | *1–5* | 8.5% (30) | |
| | *≥ 6* | 9.6% (34) | |
| | *Missing* | 0 | |
| **Classes mixing at play** | *No* | 72.8% (257) | 88% (52) |
| | *Yes: outdoors in a field or large outdoor space* | 22.4% (79) | |
| | *Yes: in the hall* | 3.7% (13) | |
| | *Missing* | (<5) | |
| **School offers breakfast club** | *No* | 36.3% (128) | 95% (56) |
| | *Yes* | 58.4% (206) | |
| | *Missing* | 5.4% (19) | |
| **School offers extra-curricular clubs** | *No* | 71.7% (253) | 91% (54) |
| | *Yes* | 26.6% (94) | |
| | *Missing* | 1.7% (6) | |
| **Teaching outdoors** | *Never/hardly ever* | 25.2% (89) | 58.6% (34) |
| | *Some of the time* | 61.8% (218) | |
| | *Most of the time* | 11.1% (39) | |
| | *Missing* | 2% (7) | |

**Table 3. Univariable (model 1) and multivariable (model 2) logistic regression models of self-reported school-based mitigation measures (survey) and school-level probability of any positive COVID-19 case in school (SAIL).**

| At least one positive COVID-19 test at school (pupils and staff) during study period (SAIL) (school-level) | | | | | | |
|---|---|---|---|---|---|---|
| **Self reported measures from survey** | | | Univariable (model 1) | | Multivariable (model 2) $R^2$ = 0.12 | |
| | | | **OR** | **95% CI** | **OR** | **95% CI** |
| **Face covering** (reference no face covering) | | *Mask* | 2.82** | 1.11 to 7.31 | 2.10* | 0.87 to 5.05 |
| | | *Visor* | 1.65 | 0.47 to 5.74 | 1.42 | 0.40 to 5.2 |
| **Keep two metres from pupils** (reference never/rarely) | | *Sometimes* | 1.01 | 0.50 to 2.02 | 0.79 | 0.36 to 1.75 |
| | | *Most of the time/ always* | 0.97 | 0.39 to 2.38 | 0.89 | 0.33 to 2.38 |
| **Keep two metres from staff** (reference never/rarely) | | *Sometimes* | 1.58 | 0.47 to 5.32 | 1.82 | 0.63 to 5.26 |
| | | *Most of the time/ always* | 2.46 | 0.76 to 7.96 | 2.85* | 0.97 to 8.37 |
| **Non-household contacts within one metre** (reference 0 contacts) | | *1–5* | 0.97 | 0.57 to 1.67 | 0.89 | 0.47 to 1.66 |
| | | *≥ 6* | 1.47 | 0.78 to 2.79 | 1.17 | 0.53 to 2.56 |
| **Non-household contacts direct** (reference 0 contacts) | | *1–5* | 2.27* | 0.98 to 5.22 | 2.89** | 1.01 to 8.31 |
| | | *≥ 6* | 1.58 | 0.86 to 2.89 | 1.70* | 0.93 to 3.10 |
| **Classes mix at play** (reference no classes mixing) | | *Yes* | 0.89 | 0.40 to 1.98 | 1.06 | 0.53 to 2.13 |
| **School offers breakfast club** (reference no breakfast club) | | *Yes* | 0.58 | 0.23 to 1.48 | 0.67 | 0.28 to 1.64 |
| **School offers extra-curricular clubs** (reference no extra-curricular clubs) | | *Yes* | 1.67 | 0.73 to 3.86 | 1.99 | 0.85 to 4.71 |
| **Teach outdoors** (reference never/hardly ever) | | *Sometimes* | 0.89 | 0.58 to 1.38 | 0.88 | 0.52 to 1.47 |
| | | *Most of the time* | 0.65 | 0.23 to 1.84 | 0.45 | 0.11 to 1.81 |

OR: Odds Ratio; 95% CI: 95% confidence intervals; p<0.05**, p<0.1*; adjusted for school size, proportion of pupils eligible for free school meals, clustered by school. Model 2 likelihood-ratio test *keep two metres from pupils* (p = 0.1) and *staff* (p = 0.03). See S3 Appendix for variable codebook.

and a trend to higher general infection (Table 4, model 3). Reporting 6 or more contacts within 1-metre was associated with higher depression (OR = 2.70, Table 6, model 8).

**Face covering.** In the univariable model there was evidence that reporting to wear a face covering was associated with an increased odds of a school-level COVID-19 case; OR = 2.82. Compared to reporting no face coverings, masks were associated with increased odds of reporting cold symptoms (multivariable model: OR = 1.98), Table 4, model 4. Reporting wearing a visor was associated with higher odds of depression (multivariable model: OR = 4.81, Table 6, model 8).

**Two metre distance from pupils or staff.** In the univariable models there were no statistically significant results to support a reduced odds for any of the outcomes when using two metre distancing. In the multivariable models we found a trend to an increased odds of a

**Table 4. Univariable (model 3) and multivariable (model 4) logistic regression models of self-reported school-based mitigation measures (survey) and individual level (school staff) self-reported cold symptoms (survey).**

| Reported cold symptoms in previous 7 days (individual level: school staff) | | | | | |
|---|---|---|---|---|---|
| **Self reported measures from survey** | | **Univariable (model 3)** | | **Multivariable (model 4) R² = 0.07** | |
| | | **OR** | **95% CI** | **OR** | **95% CI** |
| **Face covering** (reference no face covering) | *Mask* | 1.66 | 0.89 to 3.10 | 1.98** | 1.02 to 3.88 |
| | *Visor* | 2.16 | 0.76 to 6.17 | 2.35 | 0.81 to 6.86 |
| **Keep two metres from pupils** (reference never/rarely) | *Sometimes* | 0.46 | 0.16 to 0.31 | 0.50 | 0.15 to 1.62 |
| | *Most of the time/ always* | 0.79 | 0.20 to 3.14 | 0.81 | 0.22 to 2.96 |
| **Keep two metres from staff** (reference never/rarely) | *Sometimes* | 0.66 | 0.16 to 2.76 | 0.59 | 0.11 to 3.10 |
| | *Most of the time/ always* | 0.57 | 0.20 to 1.60 | 0.51 | 0.14 to 1.81 |
| **Non-household contacts within one metre** (reference 0 contacts) | *1–5* | 0.92 | 0.41 to 2.10 | 0.86 | 0.35 to 2.09 |
| | *≥ 6* | 0.85 | 0.30 to 2.46 | 0.68 | 0.16 to 2.89 |
| **Non-household contacts direct** (reference 0 contacts) | *1–5* | 2.53* | 0.85 to 7.51 | 3.09* | 0.96 to 9.93 |
| | *≥ 6* | 0.78 | 0.20 to 2.97 | 1.14 | 0.20 to 6.34 |
| **Classes mix at play** (reference no classes mixing) | *Yes* | 0.49 | 0.19 to 1.27 | 0.53 | 0.22 to 1.28 |
| **School offers breakfast club** (reference no breakfast club) | *Yes* | 0.98 | 0.46 to 2.07 | 1.15 | 0.51 to 2.58 |
| **School offers extra-curricular clubs** (reference no extra-curricular clubs) | *Yes* | 1.59 | 0.82 to 3.10 | 1.19 | 0.53 to 2.64 |
| **Teach outdoors** (reference never/hardly ever) | *Sometimes* | 0.54 | 0.23 to 1.26 | 0.60 | 0.26 to 1.36 |
| | *Most of the time* | 1.17 | 0.36 to 3.77 | 0.86 | 0.26 to 2.90 |

OR: Odds Ratio; 95% CI: 95% confidence intervals; p<0.05**, p<0.1*; adjusted for school size, proportion of pupils eligible for free school meals, clustered by school. See S3 Appendix for variable codebook.

COVID-19 positive test for the grouped exposure of staff maintaining a two metre distance from other staff most of the time/always compared to never/rarely.

**Classes mixing, breakfast club, extra-curricular clubs and teaching outdoors.** There was no significant difference in terms of infection (COVID-19 and cold) or anxiety/depression for staff in schools that allowed classes to mix, offered breakfast or extra-curricular clubs or taught outdoors most of the time.

## Qualitative results

There were 129 responses from primary school staff relating to impacts of wearing face coverings. The final categories conceptualising the impacts of wearing face coverings and frequency counts were; (i) *difficulty being heard/understood–having to talk louder* (n = 71); (ii) *difficulty*

**Table 5. Univariable (model 5) and multivariable (model 6) logistic regression models of self-reported school-based mitigation measures (survey) and individual level (school staff) moderate/severe anxiety symptoms (survey).**

| Moderate/severe anxiety (GAD-7) (individual level: school staff) | | | | | |
|---|---|---|---|---|---|
| **Self reported measures from survey** | | **Univariable (model 5)** | | **Multivariable (model 6) R² = 0.07** | |
| | | OR | 95% CI | OR | 95% CI |
| **Face covering** (reference no face covering) | *Mask* | 1.35 | 0.78 to 2.33 | 1.10 | 0.51 to 2.39 |
| | *Visor* | 2.41* | 0.87 to 6.72 | 2.58 | 0.82 to 8.08 |
| **Keep two metres from pupils** (reference never/rarely) | *Sometimes* | 0.64 | 0.31 to 1.30 | 0.62 | 0.29 to 1.35 |
| | *Most of the time/ always* | 2.12 | 0.67 to 6.68 | 2.31 | 0.72 to 7.35 |
| **Keep two metres from staff** (reference never/rarely) | *Sometimes* | 0.50 | 0.14 to 1.76 | 0.53 | 0.14 to 2.06 |
| | *Most of the time/ always* | 0.63 | 0.21 to 1.91 | 0.77 | 0.21 to 2.76 |
| **Non-household contacts within one metre** (reference 0 contacts) | *1–5* | 0.90 | 0.42 to 1.89 | 0.85 | 0.39 to 1.87 |
| | *≥ 6* | 1.31 | 0.59 to 2.88 | 1.41 | 0.64 to 3.08 |
| **Non-household contacts direct** (reference 0 contacts) | *1–5* | 0.58 | 0.18 to 1.92 | 0.62 | 0.18 to 2.13 |
| | *≥ 6* | 1.59 | 0.47 to 5.34 | 2.03 | 0.55 to 7.52 |
| **Classes mix at play** (reference no classes mixing) | *Yes* | 0.99 | 0.49 to 1.99 | 0.93 | 0.43 to 2.02 |
| **School offers breakfast club** (reference no breakfast club) | *Yes* | 0.70 | 0.38 to 1.27 | 0.77 | 0.38 to 1.55 |
| **School offers extra-curricular clubs** (reference no extra-curricular clubs) | *Yes* | 1.22 | 0.50 to 2.94 | 1.25 | 0.44 to 3.56 |
| **Teach outdoors** (reference never/hardly ever) | *Sometimes* | 0.65 | 0.34 to 1.22 | 0.62 | 0.31 to 1.25 |
| | *Most of the time* | 0.70 | 0.26 to 1.87 | 0.70 | 0.25 to 1.94 |

OR: Odds Ratio; 95% CI: 95% confidence intervals; p<0.05**, p<0.1*; adjusted for school size, proportion of pupils eligible for free school meals, clustered by school. See S3 Appendix for variable codebook.

*understanding body language/facial expressions* (n = 25); *(iii) physical impacts of wearing a face covering including impacts on health and vision* (n = 22); *(iv) social/emotional impacts affecting relationships with pupils* (n = 12); *(v) challenges for pupils with additional learning needs and English as an additional language* (n = 9); and *(vi) impact on teaching phonics* (n = 6). In some instances, quotes were coded within multiple categories due to the open-ended nature of the survey question allowing long text responses. A summary of each category is discussed below and additional key quotes are presented in S5 Appendix.

**Difficulty being heard/understood—Having to talk louder.** The most frequent impact of wearing face coverings was the challenge of being heard or understood by pupils. This required staff to have to stand closer to pupils and to raise their voice to be heard. School staff reported that they found it difficult to hear others wearing a mask, and this was a particular issue for staff with hearing problems.

**Table 6. Univariable (model 7) and multivariable (model 8) logistic regression models of self-reported school-based mitigation measures (survey) and individual level (school staff) moderate/severe depressive symptoms (survey).**

| Moderate/severe depression (PHQ-9) (individual level: school staff) | | | | | | |
|---|---|---|---|---|---|---|
| **Self reported measures from survey** | | | **Univariable (model 7)** | | **Multivariable (model 8) R² = 0.07** | |
| | | | **OR** | **95% CI** | **OR** | **95% CI** |
| **Face covering** (reference no face covering) | | *Mask* | 1.78 | 0.93 to 3.42 | 1.70 | 0.83 to 3.48 |
| | | *Visor* | 3.38** | 1.31 to 8.77 | 4.81** | 1.52 to 15.22 |
| **Keep two metres from pupils** (reference never/rarely) | | *Sometimes* | 1.03 | 0.50 to 2.15 | 0.97 | 0.40 to 2.36 |
| | | *Most of the time/ always* | 1.18 | 0.50 to 2.78 | 1.95 | 0.61 to 6.21 |
| **Keep two metres from staff** (reference never/rarely) | | *Sometimes* | 1.26 | 0.29 to 5.36 | 0.68 | 0.13 to 3.48 |
| | | *Most of the time/ always* | 1.05 | 0.28 to 3.97 | 0.73 | 0.16 to 3.26 |
| **Non-household contacts within one metre** (reference 0 contacts) | | *1–5* | 1.44 | 0.73 to 2.84 | 1.88 | 0.74 to 4.75 |
| | | *≥ 6* | 1.65 | 0.76 to 3.59 | 2.70** | 1.11 to 6.56 |
| **Non-household contacts direct** (reference 0 contacts) | | *1–5* | 1.12 | 0.45 to 2.77 | 0.90 | 0.27 to 3.00 |
| | | *≥ 6* | 1.28 | 0.45 to 3.68 | 1.17 | 0.35 to 3.98 |
| **Classes mix at play** (reference no classes mixing) | | *Yes* | 0.82 | 0.41 to 1.64 | 0.82 | 0.30 to 2.22 |
| **School offers breakfast club** (reference no breakfast club) | | *Yes* | 0.73 | 0.40 to 1.34 | 0.89 | 0.32 to 2.44 |
| **School offers extra-curricular clubs** (reference no extra-curricular clubs) | | *Yes* | 1.03 | 0.35 to 3.05 | 0.87 | 0.24 to 3.21 |
| **Teach outdoors** (reference never/hardly ever) | | *Sometimes* | 0.86 | 0.40 to 1.84 | 0.75 | 0.30 to 1.91 |
| | | *Most of the time* | 1.84 | 0.56 to 6.06 | 1.59 | 0.39 to 6.50 |

OR: Odds Ratio; 95% CI: 95% confidence intervals; p<0.05**, p<0.1*; adjusted for school size, proportion of pupils eligible for free school meals, clustered by school. See S3 Appendix for variable codebook.

> "*Pupils can't always hear me so I have to lift the visor. . .when two meters away and talk louder when I am closer to support pupils*" (teaching assistant)

**Difficulty understanding body language/facial expressions.** School staff noted a challenge for pupils in understanding the body language or interpreting facial expressions of adults. This impacted staff in this study to communicate with children and was particularly challenging for younger pupils.

> "*I find it extremely difficult to wear a mask/visor whilst teaching. They are young children and need to see facial expressions. It also affects my hearing and their ability to hear me clearly*" (teacher)

**Physical impacts of wearing a face covering including impacts on health and vision.**
School staff reported physical impacts and negative complaints including feelings of discomfort. Other common negative effects included their vision, headaches and sore throat. Underlying medical conditions including asthma contributed to challenges experienced by staff with perceived restrictions to breathing.

"*Visors are really difficult, they make me feel enclosed and stressed. The children cannot hear me and the vision is not brilliant either*" (teacher)

**Social/emotional impacts affecting relationships with pupils.** Those that wore a face covering and particularly mask use commented on the emotional impact of children not being able to interpret emotions. Staff perceived that this had an impact on their relationship with pupils.

"*Yes, the children would not be able to see my expression, if they are upset they wouldn't be able to see my reaction or compassion*" (teaching assistant)

**Challenges for pupils with additional learning needs and English as an additional language.** Additional challenges were presented with supporting children with additional learning needs (ALN) or English as an additional language (EAL), with mask use impacting communication and language development.

"*Yes, it's affecting my teaching. I work with pupils who are learning English as an additional language and they ideally need to be able to see my facial expressions and lip movements in order to help them understand and develop the language themselves*" (teacher)

**Impact on teaching phonics.** School staff specifically made references to teaching phonics, including the challenges of teaching reading, writing and language skills. Some felt that face masks restricted modelling of words and demonstrating pronunciation.

"*Pupils in my class have low language development. They need to see my mouth to support the modelling of words and phonics. Greater effort in delivering modelled speech can become tiring very quickly*" (teacher)

## Discussion

This study aims to examine the association of different school-based mitigation measures reported by primary school staff between October to December 2020 on the likelihood of any school-level COVID-19 infection (pupils and staff) at the linked school during this period. This study also examined the association of these measures with individual-level self-reported infection (cold symptoms), anxiety and depression of school staff. Findings suggest that reporting more direct non-household contacts was associated with higher odds of COVID-19 at the school level, and a trend towards self-reported infection. Reporting six or more non-house contacts within 1-metre was also associated with higher depression in school staff. We found no evidence that reporting wearing face coverings or maintaining a two metre distance from pupils or other staff during the study period was associated with lower odds of COVID-19 in the linked school setting.

Whilst this observational study offers a real-world evaluation of the school setting, findings highlight the challenge for staff in implementing and adhering to school guidelines. This study

assumes that reported measures were in place for the duration of the study period in line with operational guidance issued to schools at that time. However, changes in day-to-day school practice brings methodological challenges of evaluating compliance with and effectiveness of national-level guidance. Our findings of within-school agreement suggests some measures are implemented at a school-level (face coverings, mixing classes at play, breakfast and extra-curricular clubs). In comparison, agreement of other measures (number of contacts, maintaining two metre distance from pupils and staff and teaching outdoors) suggest individual-level influences of adherence to measures, reflecting the challenge of implementing generic guidance in a dynamic school environment.

The finding that reduced contacts may be protective at the school-level is important within the contexts of different settings where the implementation and adherence to different blanket mitigation measures varies. Specifically, this study finds an association between the number of direct physical contacts and increased likelihood of COVID-19 school infections. It is well established that contact patterns of close proximity, prolonged contact and contact frequency are strongly associated with increased risk of transmission [33]. Our finding is consistent with the evidence base regarding contact patterns where reducing number of contacts is associated with a reduction in the basic reproduction number ($R_0$) [34]. A crossover between COVID-19 and common cold symptoms has been established [25], and the current study also found an association between direct physical contacts and self-reported cold symptoms. As this study suggests variation of school-based mitigation measures between and within-schools, encouraging individual behaviours of school staff such as reducing direct contacts may be of benefit in reducing transmission of COVID-19 or general infection in the school setting.

Relating to proximity, qualitative findings from this study suggest challenges for staff wearing face coverings including pupils having difficulty hearing and understanding, and this required them to talk louder or move physically closer to pupils to be heard. Research demonstrates that people speak louder when wearing masks [35]. Staff also noted that pupils were unable to interpret facial expressions or emotions, impacting their relationship with pupils and children's perception of compassionate emotions conveyed by staff. Challenges were cited for ALN or EAL pupils particularly regarding speech and language development. As facial expressions and gestures are largely responsible for verbal, non-verbal and emotional face-to-face communication, face masks may hinder interpersonal communication with pupils [36].

Type of face mask was not captured in this study (e.g. medical/non-medical grade). Guidance to primary schools during the study period (autumn term 2020) did not enforce medical-grade face coverings [9,37]. The type of face covering worn by staff in this study may include cloth masks which have been found to increase respiratory infection risk due to moisture retention, reuse and poor filtration [38]. This may explain individual-level findings that staff wearing face masks had higher odds of reporting cold symptoms in the previous seven days. In the context of SARS-CoV-2 transmission, the main purpose of face coverings is to prevent onward transmission to others as opposed to protecting the individual wearing the face covering [39]. A systematic review and meta-analysis showed a reduction in COVID-19 incidence with mask wearing, though type of fask mask, compliance and frequency of use were not captured [40]. It is important to note the many confounding variables of face covering usage that were not measured in this study. This includes background prevalence in the area which may influence wearing face coverings. Evidence suggests that mandating face covering use alone may not increase usage and thus, individual behaviours and other influences are likely to play a role in face covering behaviour [41]. In addition, this observational study assumes reported mitigation measures were in effect for the duration of the study period. It is possible that

reverse causality occurred, that is a school staff practitioner may have chosen to wear a face covering following the onset of common cold symptoms.

The use of visors was associated with higher anxiety/depression for staff in this study. Impacts on teacher wellbeing have been highlighted in previous research by HAPPEN during school closures and the phased reopening of schools in the summer term of 2020, with primary school staff advocating for their wellbeing to be prioritised [42]. This is important as teacher wellbeing is associated with academic achievement [13]. School staff in the current study also commented on the physical impacts of wearing face coverings, including negatively affecting their vision, causing headaches and breathing difficulties. Qualitative research exploring face covering behaviour has highlighted the wide range of motivations, including individual and community protection, and barriers such as physical challenges and discomfort [43]. It is possible that the physical discomforts expressed by staff in this study influence face covering behaviour.

We found no evidence in this study that maintaining a two metre distance from pupils reduces the odds of a COVID-19 school-level incident. However, few staff were able to achieve this. Research examining the implementation of preventive school-based measures in primary schools in England highlights the challenge of maintaining physical distancing from pupils and the negative impact of distancing measures on teaching including teaching letter formation [10]. This finding is mirrored in the current study, with specific references to the challenges of teaching phonics and those discussed previously. The potential consequences of failing to address these pedagogical impacts include pupils falling further behind in their learning [44].

This study did not find evidence of higher odds of COVID-19 school incidents where children from different classes mix, including breakfast club, extra-curricular clubs and mixing different classes at playtime. School provision during the COVID-19 pandemic encompasses balancing transmission risks against the benefits for children's social and emotional development, wider skill development, educational attainment and reducing inequalities. The COVID-19 pandemic has exacerbated pre-existing inequalities including food insecurity, child poverty and child hunger [45,46] which negatively impact educational attainment [47]. Provision such as breakfast clubs that address socio-economic inequalities are of great public health, education and economic importance and this was reflected in guidance at the time of the study encouraging breakfast clubs [9].

The World Health Organization (WHO), UNICEF and UNESCO recently updated advice to policymakers and educators, issuing a set of risk-based considerations regarding school provision since reopening during the COVID-19 pandemic [48]. Whilst the principles aim to prevent and minimise transmission risks within the school setting, the WHO advocate that at the forefront, educational settings should prioritise "*the continuity of education for children for their overall well-being, health and safety*", the "*social learning and development of children*" and to consider implications of decisions on school staff. Findings from this study highlight the challenges of evaluating the implementation of guidance and the variation in implementation at an individual and school-level. Governments continually review available evidence to inform risk-based approaches to education delivery that safeguard children's learning, health and wellbeing and support school staff. This must consider the risk of transmission in addition to the impacts on pupils, teachers and senior school leaders. Finally, both the Welsh and UK governments have recently announced plans to reverse some of these guidelines for schools in the upcoming 2021/22 academic year starting in September 2021. This includes the removal of isolation policies for children in close contact with confirmed cases, removing the use of school 'bubbles' to segregate year groups, and face coverings will no longer be recommended.

## Strengths and limitations

All primary schools in Wales (n = 1,203) were contacted however the findings in this study are a convenience sample, only representing those that participated and may not be representative of non-participating schools. A range of school-based measures have been implemented and the findings in this study may not encapsulate all approaches. School-based mitigation measures included in analyses were obtained from a self-report survey and may result in recall bias. This is an observational study and so cannot show cause and effect. As with all observational studies, unmeasured confounders and reverse causality may influence findings, e.g., face covering usage may increase due to a previous COVID-19 case in the school, higher community prevalence and individual behaviours. Thus, face covering use and future COVID-19 cases may be linked by an unmeasured confounder. This study assumed that reported measures were in effect for the duration of the period of study based on national-level guidance issued to schools by the Welsh Government at the start of the autumn term 2020. It is possible that within-schools' day to day practice varied. Despite this, the sample consists of a range of primary school staff including headteachers, teachers and support staff working in schools in 15 of 22 local authorities in Wales, of varying school size and ranges of pupils eligible for free school meals. This study was able to examine all COVID-19 PCR test results in Wales and link these to the relevant school setting and so gives an objective assessment of the association of self-reported adherence to mitigation measures and COVID-19 test positive cases.

## Conclusions

Implementation of COVID-19 mitigation measures was variable and challenging in primary schools in Wales. This study did find evidence that reducing the number of direct non-household contacts is associated with lower risk of COVID-19 in the school and general infection for the individual. This study did not find evidence that face coverings, two metre social distancing, stopping children mixing or removing breakfast clubs are associated with fewer COVID-19 cases in the school or with lower general infection rates and did find evidence that these measures can affect teaching quality.

## Supporting information

**S1 Appendix. RECORD statement.**
(DOCX)

**S2 Appendix. Full survey copy.**
(DOCX)

**S3 Appendix. Staff survey variable codebook.**
(DOCX)

**S4 Appendix. Sensitivity analyses.**
(DOCX)

**S5 Appendix. Additional qualitative quotes.**
(DOCX)

## Acknowledgments

The authors would like to thank all participating primary schools and school staff that took part in this study. Infrastructure support was received by the National Centre for Population Health and Wellbeing Research through the HAPPEN network. This study makes use of

anonymised data held in the Secure Anonymised Information Linkage (SAIL) Databank. We would like to acknowledge all the data providers who make anonymised data available for research.

## Author Contributions

**Conceptualization:** Emily Marchant, Sinead Brophy.

**Data curation:** Emily Marchant, Jonathan Kennedy.

**Formal analysis:** Emily Marchant.

**Funding acquisition:** Ronan A. Lyons.

**Investigation:** Emily Marchant.

**Methodology:** Emily Marchant.

**Project administration:** Emily Marchant.

**Resources:** Emily Marchant.

**Supervision:** Lucy Griffiths, Tom Crick, Sinead Brophy.

**Validation:** Joe Hollinghurst.

**Writing – original draft:** Emily Marchant.

**Writing – review & editing:** Emily Marchant, Lucy Griffiths, Tom Crick, Richard Fry, Michaela James, Laura Cowley, Hoda Abbasizanjani, Fatemeh Torabi, Daniel A. Thompson, Ashley Akbari, Michael B. Gravenor, Ronan A. Lyons, Sinead Brophy.

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
