## [Decision Letter · Decision Letter 0]

1 Nov 2021

PONE-D-21-27732COVID-19 mitigation measures in primary schools and association with infection and school staff wellbeing: an observational survey linked with routine data in Wales, UKPLOS ONE

Dear Dr. Marchant,

Thank you for submitting your manuscript to PLOS ONE. After careful consideration, we feel that it has merit but does not fully meet PLOS ONE’s publication criteria as it currently stands. Therefore, we invite you to submit a revised version of the manuscript that addresses the points raised during the review process.

We look forward to receiving your revised manuscript.

Kind regards,

Amitava Mukherjee, ME, Ph.D.

Academic Editor

PLOS ONE

Journal Requirements:

Reviewers' comments:

Reviewer's Responses to Questions

**Comments to the Author**

1. Is the manuscript technically sound, and do the data support the conclusions?

Reviewer #1: Partly

2. Has the statistical analysis been performed appropriately and rigorously? 

Reviewer #1: I Don't Know

3. Have the authors made all data underlying the findings in their manuscript fully available?

Reviewer #1: Yes

4. Is the manuscript presented in an intelligible fashion and written in standard English?

Reviewer #1: Yes

5. Review Comments to the Author

Reviewer #1: Thank you for the opportunity to review this paper. COVID-19 mitigation measures in primary schools and associations with infection is an interesting and important topic. The low risk of severe disease and death in children has led to debates on whether school closures and mitigation measures affecting learning are justified.

Despite the density of data and the important topic, my overall impression is that the paper would benefit from being more focused. There are four separate outcomes for associations with staff self-reported mitigation measures; (i) school-level positive COVID-19 cases (primary outcome), (ii) self-reported cold symptoms, (iii) moderate/severe anxiety and (iv) depressive symptoms, and adding qualitative data to this.

Comments:

• The univariable and multivariable OR were presented separately rather than in the same table for the same outcome. I had to scroll back and forth several times to follow the flow. As a reader I would prefer that univariable and multivariable results are presented in the same table with one table per outcome.

• Numbers already presented in the tables were repeated numerically in the text, adding to information overload.

• The questions was phrased with multiple alternatives (see below) whereas the analyses are done separately as dichotomous exposures. It is not clear whether the research question is relative or related to the exposure as whole. I would have expected that Keeping 2 metres from STAFF was treated as one variable (exposure as whole), and that this variable should be tested for significance before interpreting the significance of the subvariables (rarely, some of the time, most of the time, always) I think authors need to clarify this.

Keep 2 metres from STAFF 34. How often are you able to keep 2 metres away from other members of staff during the teaching day? i) Never

ii) Rarely

iii) Some of the time

iv) Most of the time

v) Always

• It is not clear whether the authors corrected for multiple comparisons, which would be relevant if the outcome is treated as categorical variables. The outcome is not frequent and most associations are not significant.

• The authors refer to two different levels of significance <0.5 and <0.1, which is fine. However, these levels are only presented with bold or italic OR and CI in the tables. It would be easier for the reader if the p-value information was presented numerically, or by one and two asterix depending on the level of significance.

• Adding qualitative data to the manuscript is interesting and important. However, the analysis is drowning in all the other information and would deserve better.

• Self-reported cold symptoms in the previous seven days is included as a separate outcome (proxy for infection risk). However, it is not clear how this should be interpreted and what it adds to the research question.

• In the discussion they find that staff wearing face-masks had higher odds of reporting cold symptoms in the previous seven days. It is not clear whether they were more likely to use masks due to cold-symptoms, rather than vice-versa.

• The authors suggests that asymptomatic transmission from children could be an explanation. However, asymptomatic transmission from children in school settings is not very common.

6. PLOS authors have the option to publish the peer review history of their article (what does this mean?). If published, this will include your full peer review and any attached files.

Reviewer #1: No

---

## [Author Response · Author response to Decision Letter 0]

4 Jan 2022

Reviewer #1: Thank you for the opportunity to review this paper. COVID-19 mitigation measures in primary schools and associations with infection is an interesting and important topic. The low risk of severe disease and death in children has led to debates on whether school closures and mitigation measures affecting learning are justified.

Despite the density of data and the important topic, my overall impression is that the paper would benefit from being more focused. There are four separate outcomes for associations with staff self-reported mitigation measures; (i) school-level positive COVID-19 cases (primary outcome), (ii) self-reported cold symptoms, (iii) moderate/severe anxiety and (iv) depressive symptoms, and adding qualitative data to this.

We would like to thank you for taking the time to read our manuscript and to provide your comments. The manuscript covers two themes: infection (COVID & cold like symptoms), and wellbeing (anxiety and depression) with complementary qualitative data to give context to quantitative findings. The impacts of the COVID-19 pandemic are very much on risk of infection and the impact of mitigation measures on mental health. We feel these aspects need to be addressed together as one impacts on each other. Therefore, whilst we appreciate and agree that this may feel like a data dense manuscript, we believe the risk of infection has to be discussed alongside the impact to mental health. We also believe that the qualitative context is very important in understanding the full picture, and adds a significant contribution to the literature. 

Please find section, page and line numbers presenting revisions below, with manuscript text in italic.

Comments:

• The univariable and multivariable OR were presented separately rather than in the same table for the same outcome. I had to scroll back and forth several times to follow the flow. As a reader I would prefer that univariable and multivariable results are presented in the same table with one table per outcome.

Thank you for raising this point. We agree that presenting the univariable and multivariable analyses in the same table ensures clarity for the reader. We have addressed this as suggested, presenting each outcome within separate tables with univariable and multivariable analyses. These tables and models have been renumbered as below:

Quantitative results

Page 15-21

Table 3: School-level probability of any positive COVID-19 case in school (model 1: univariable, model 2: multivariable) 

Table 4: Individual level (school staff) self-reported cold symptoms (model 3: univariable, model 4: multivariable)

Table 5: Individual level (school staff) moderate/severe anxiety symptoms (model 5: univariable, model 6 multivariable)

Table 6: Individual level (school staff) moderate/severe depressive symptoms (model 7: univariable, model 8 multivariable)

• Numbers already presented in the tables were repeated numerically in the text, adding to information overload.

We appreciate you drawing attention to the presentation of our results and we agree that repeating the results numerically in the text may overload the reader. To address your comment, we have removed the majority of numerical results. We feel that in instances of statistically significant results at the 5% level, stating the Odds Ratio is useful for the reader. However, we have removed the 95% CIs to ensure results are concise and readable. This style of presentation of results can also be found in other Covid published research in PLOS ONE. 

Quantitative results

Page 14, line 321-325

In the multivariable models, compared to reporting 0 contacts, reporting more non-household direct contacts was associated with higher odds of COVID-19 at the school level (1-5 contacts OR = 2.89, Table 3, model 1), and a trend to higher general infection (Table 4, model 3). Reporting 6 or more contacts within 1-metre was associated with higher depression (OR = 2.70, Table 6, model 8).

• The questions was phrased with multiple alternatives (see below) whereas the analyses are done separately as dichotomous exposures. It is not clear whether the research question is relative or related to the exposure as whole. I would have expected that Keeping 2 metres from STAFF was treated as one variable (exposure as whole), and that this variable should be tested for significance before interpreting the significance of the subvariables (rarely, some of the time, most of the time, always) I think authors need to clarify this.

Keep 2 metres from STAFF 34. How often are you able to keep 2 metres away from other members of staff during the teaching day? i) Never, ii) Rarely, iii) Some of the time, iv) Most of the time, v) Always

• It is not clear whether the authors corrected for multiple comparisons, which would be relevant if the outcome is treated as categorical variables. The outcome is not frequent and most associations are not significant.

Thank you for raising this point regarding some survey items and associated response categories and exposure coding. We assigned ordinal categories for the purpose of analyses to some items, including the item you have highlighted (staff social distancing). We treated these variables as categorical in the analyses, with the reference category indicated in the results tables. An example is demonstrated below and presented in S3 Appendix: 

Exposures Survey item Survey responses categories Coding for analyses 

Keep two metres from STAFF 34. How often are you able to keep 2 metres away from other members of staff during the teaching day? i) Never

ii) Rarely

iii) Some of the time

iv) Most of the time

v) Always Ordinal:

- Never/rarely (i, ii)

- Some of the time (iii)

- Most of the time/always (iv, v)

Example taken from S3 Appendix

Analyses with ordinal responses used the grouped never/rarely category as the reference group. 

We did not conduct sub-variable analyses but included the distributions of response categories in Table 2. To improve the clarity for readers and address the points you have raised, we have updated Table 2 to reflect the exposure categories used within analyses. For example:

Survey item Response % (n) % (n) of schools with �80% agreement of responses (for school-level outcome)

Keep two metres from pupils Never/rarely 67.1% (237) 61% (36)

 Sometimes 23.5% (83) 

 Most of the time/always 7.9% (28) 

 Missing 1.4% (5) 

Example taken from table 2 (page 12-13)

We have also updated the layout and presentation of tables 3 to 6 to aid the interpretation of tables. For example: 

Self reported measures from survey Univariable (model 1) Multivariable (model 2)

 OR 95% CI OR 95% CI

Keep two metres from staff

(reference never/rarely) Sometimes 1.58 0.47 to 5.32 1.82 0.63 to 5.26

 Most of the time/always 2.46 0.76 to 7.96 2.85* 0.97 to 8.37

Example taken from table 3 (page 15-16)

We have presented a full breakdown of exposures included within analyses, survey item, response and coding in S3 Appendix. However, we agree that further clarity is required within the manuscript. To address this within the manuscript text, we have included additional explanations as below: 

School staff survey and linked data:

Page 7, line 160-162

A full copy of the survey is presented in S2 Appendix, and detail regarding survey item, item response categories and item coding for analyses are presented in S3 Appendix. 

Quantitative analysis

Page 8-9, line 193-206

All exposure measures relating to government guidance were captured through self-report by school staff via the online survey and were analysed in individual models (univariable) and then in a combined model (multivariable). For the purpose of analyses, items with multiple category responses or continuous numerical values were assigned ordinal categories. For example survey response categories for keep two metres from pupils/staff included i) never, ii) rarely, iii) some of the time, iv) most of the time, v) always, with combined ordinal categories for analyses of i) never/rarely, ii) some of the time, iii) most of the time/always. Further detail of exposures for all survey items within analyses including possible response category, grouping and coding can be found in S3 Appendix. This study assumed self-reported mitigation measures to be in effect for the duration of the study period based on operational guidance issued to schools at the time of the study [9].

We have also amended the column title in S3 Appendix to Coding for analyses. 

Finally, we have addressed this within the results write up:

Quantitative results

Page 14, line 336-339

In the multivariable models we found a trend to an increased odds of a COVID-19 positive test for the grouped exposure of staff maintaining a 2-metre distance from other staff most of the time/always compared to never/rarely

• The authors refer to two different levels of significance <0.5 and <0.1, which is fine. However, these levels are only presented with bold or italic OR and CI in the tables. It would be easier for the reader if the p-value information was presented numerically, or by one and two asterix depending on the level of significance.

We appreciate that bold or italic is unclear for the reader. To address this, we have amended tables 3-6 (pages 15-21) presenting univariable and multivariable analyses with one (p<0.1) and two (p<0.05) asterisks. 

For example:

At least one school positive COVID-19 test (pupils and staff) during study period (SAIL) (school-level)

Self reported measures from survey Univariable (model 1) Multivariable (model 2)

 OR 95% CI OR 95% CI

Face covering 

(reference no face covering)

 Mask 2.82** 1.11 to 7.31 2.10* 0.87 to 5.05

 Visor 1.65 0.47 to 5.74 1.42 0.40 to 5.2

Taken from table 2

• Adding qualitative data to the manuscript is interesting and important. However, the analysis is drowning in all the other information and would deserve better.

We agree with the importance of providing complementary qualitative data within our manuscript, and believe it offers a rich perspective of school staff that have been required to adapt their teaching practice and adhere to a range of school-based mitigation measures. Qualitative research can offer a meaningful contribution to shaping policy and practice. In the case of this study, we explored staff perspectives of the impacts of face coverings to further explain the quantitative findings. 

As this is a secondary outcome of the manuscript we were conscious to ensure that the qualitative results were discussed succinctly and clearly. We also felt it was important to include one example verbatim quote to represent each theme, with additional verbatim quotes available for the reader in S5 Appendix. Whilst it would be possible to dedicate an entire qualitative paper to these findings, we feel they complement the quantitative data and offer potential mechanisms to explain these findings, as outlined in the discussion. 

Furthermore, the immediacy of the COVID-19 pandemic requires rapid research to provide evidence informing emerging policy and practice, which we believe we should include all findings in this manuscript to ensure research findings are delivered in a timely way so that they can be used to inform practice in schools. 

Introduction

Page 5, line 102-104

Secondary qualitative data exploring the impacts of wearing face coverings are also presented to complement quantitative findings.

• Self-reported cold symptoms in the previous seven days is included as a separate outcome (proxy for infection risk). However, it is not clear how this should be interpreted and what it adds to the research question.

Thank you for raising this point. We agree that additional clarity is required to explain the inclusion of this outcome and have addressed this within the manuscript as listed below. We have used self-reported cold symptoms in the previous seven days as a proxy for infection (either COVID-19 or general infection). Evidence from the UK ZOE COVID study shows the crossover of symptoms between the common cold and COVID-19, particularly following two vaccine doses. It is also possible that Covid cases in the school (either pupil or staff) were not detected, including asymptomatic transmission or not being tested. Therefore, reporting cold symptoms indicates that transmission of either COVID-19 or other general infections is occurring. In the case of the school setting, this could suggest that some school-based mitigation measures may not be effective in preventing transmission, or are not being adhered to. In addition, if staff reported general cold or other viral infection symptoms, it is possible that this was transmitted within the school environment and would suggest they could also be exposed to COVID-19 or be transmitting an infection onwards to others. 

We have addressed this as outlined below, including the addition of a reference to support the crossover between COVID and cold symptoms.

Quantitative analysis

Page 8, line 180-183

Secondary binary outcomes investigated at an individual level captured by the online survey included self-reported cold symptoms in the previous seven days as a proxy of infection risk as evidence suggests a crossover of symptoms between COVID-19 and the common cold [25], and moderate/severe anxiety (GAD-7) and moderate/severe depression (PHQ-9).

Discussion

Page 26, line 523-525

A crossover between COVID-19 and common cold symptoms has been established [25], and the current study also found an association between direct physical contacts and self-reported cold symptoms. As this study suggests variation of school-based mitigation measures between and within-schools, encouraging individual behaviours of school staff such as reducing direct contacts may be of benefit in reducing transmission of COVID-19 or general infection in the school setting.

• In the discussion they find that staff wearing face-masks had higher odds of reporting cold symptoms in the previous seven days. It is not clear whether they were more likely to use masks due to cold-symptoms, rather than vice-versa.

Thank you for raising this important point. Our study assumes that reported measures including the use of face coverings were in effect for the duration of the study period. We agree that reverse causality may be occurring in some instances. Whilst we acknowledged this within the strengths and limitations section (below), we have addressed this by also outlining this within the discussion:

Discussion 

Page 27, line 551-555

In addition, this observational study assumes reported mitigation measures were in effect for the duration of the study period. It is possible that reverse causality occurred, that is a school staff practitioner may have chosen to wear a face covering following the onset of common cold symptoms. 

Strengths and limitations 

Page 30, line 615-618

As with all observational studies, unmeasured confounders and reverse causality may influence findings, e.g., face covering usage may increase due to a previous COVID-19 case in the school, higher community prevalence and individual behaviours. Thus, face covering use and future COVID-19 cases may be linked by an unmeasured confounder.

• The authors suggests that asymptomatic transmission from children could be an explanation. However, asymptomatic transmission from children in school settings is not very common.

Thank you for drawing our attention to this. We have removed the sentence that suggests asymptomatic transmission could be occurring as it is not possible to ascertain this from the current study.

---

## [Decision Letter · Decision Letter 1]

26 Jan 2022

PONE-D-21-27732R1COVID-19 mitigation measures in primary schools and association with infection and school staff wellbeing: an observational survey linked with routine data in Wales, UKPLOS ONE

Dear Dr. Marchant,

Thank you for submitting your manuscript to PLOS ONE. After careful consideration, we feel that it has merit but does not fully meet PLOS ONE’s publication criteria as it currently stands. Therefore, we invite you to submit a revised version of the manuscript that addresses the points raised during the review process.

We look forward to receiving your revised manuscript.

Kind regards,

Amitava Mukherjee, ME, Ph.D.

Academic Editor

PLOS ONE

Journal Requirements:

Reviewers' comments:

Reviewer's Responses to Questions

**Comments to the Author**

1. If the authors have adequately addressed your comments raised in a previous round of review and you feel that this manuscript is now acceptable for publication, you may indicate that here to bypass the “Comments to the Author” section, enter your conflict of interest statement in the “Confidential to Editor” section, and submit your "Accept" recommendation.

Reviewer #1: (No Response)

2. Is the manuscript technically sound, and do the data support the conclusions?

Reviewer #1: Partly

3. Has the statistical analysis been performed appropriately and rigorously? 

Reviewer #1: No

4. Have the authors made all data underlying the findings in their manuscript fully available?

Reviewer #1: Yes

5. Is the manuscript presented in an intelligible fashion and written in standard English?

Reviewer #1: Yes

6. Review Comments to the Author

Reviewer #1: Thank you for the invitation to review the revised version of the manuscript.

The authors have made substantial revisions responding to previous comments, and the flow and readability of the paper is much improved.

I believe I was not sufficiently clear in one of my previous comments regarding the multivariable analysis. My previous comment was:

The questions was phrased with multiple alternatives (see below) whereas the analyses are done separately as dichotomous exposures. It is not clear whether the research question is relative or related to the exposure as whole. I would have expected that Keeping 2 metres from STAFF was treated as one variable (exposure as whole), and that this variable should be tested for significance before interpreting the significance of the subvariables (rarely, some of the time, most of the time, always) I think authors need to clarify this.

This was not related to ordinal data, but to the validity of the analysis. Before doing pairwise comparisons of separate values within a variable (i.e. testing “sometimes” versus “never/rarely”) the variable “keeping 2 meters distance” should be tested for significance. The p-value for the whole variable can be calculated in two ways:

1. A likelihood ratio test comparing a model containing the variable vs a model not containing the variable

2. A wald test that tests the joint significance of beta1 = beta2 = 0 (where beta1 is the coefficient for sometimes and beta2 is the coefficient for most of the time/always)

I believe pairwise comparisons (i.e. testing “sometimes” versus “never/rarely”) should be limited to significant variables to make sense, and the the p-value for the whole variable should be presented in the table alongside the pairwise comparisons.

I have no further comments

7. PLOS authors have the option to publish the peer review history of their article (what does this mean?). If published, this will include your full peer review and any attached files.

Reviewer #1: No

---

## [Author Response · Author response to Decision Letter 1]

28 Jan 2022

Reviewer #1:

The questions was phrased with multiple alternatives (see below) whereas the analyses are done separately as dichotomous exposures. It is not clear whether the research question is relative or related to the exposure as whole. I would have expected that Keeping 2 metres from STAFF was treated as one variable (exposure as whole), and that this variable should be tested for significance before interpreting the significance of the subvariables (rarely, some of the time, most of the time, always) I think authors need to clarify this.

This was not related to ordinal data, but to the validity of the analysis. Before doing pairwise comparisons of separate values within a variable (i.e. testing “sometimes” versus “never/rarely”) the variable “keeping 2 meters distance” should be tested for significance. The p-value for the whole variable can be calculated in two ways:

1. A likelihood ratio test comparing a model containing the variable vs a model not containing the variable

2. A wald test that tests the joint significance of beta1 = beta2 = 0 (where beta1 is the coefficient for sometimes and beta2 is the coefficient for most of the time/always)

I believe pairwise comparisons (i.e. testing “sometimes” versus “never/rarely”) should be limited to significant variables to make sense, and the the p-value for the whole variable should be presented in the table alongside the pairwise comparisons.

I have no further comments

On my behalf of the co-authors, we would like to thank you for taking the time to read our revised manuscript. We are glad that you are satisfied that we have addressed the useful comments provided and that this has improved the flow and readability of the paper. 

Thank you for drawing further attention and providing additional clarity regarding your previous comment about the multivariable analysis. Further to the changes in revision one, we have addressed this point as you have suggested by performing a likelihood ratio test for the variable keep two metres as a whole variable i.e. survey response categories. We have assessed the goodness of fit of two alternative models including and excluding this whole variable. The output from this test shows a significantly improved difference with the inclusion of the keep two metres from staff variable (p=0.03), a variable that we found to be associated with increased likelihood of COVID-19 positive test in the school. For keep two metres from pupils, the output presents a likelihood ratio of p=0.1 (borderline significant at the 10% level), though we did not report on this variable as we found no association with outcomes in multivariable models. To ease interpretation of Odds Ratios, we created ordinal categories relating to level of exposure. 

We have addressed this within the manuscript, including stating within the methods, results and in the Table 3 footnote. Please find below further information of these revisions:

Quantitative analysis, page 8-9, line 191-198.

Items with multiple category responses or continuous numerical values were assigned ordinal categories to ease interpretation. For example survey response categories for keep two metres from pupils/staff included i) never, ii) rarely, iii) some of the time, iv) most of the time, v) always, with combined ordinal categories for analyses of i) never/rarely, ii) some of the time, iii) most of the time/always. For these variables, likelihood-ratio tests of variables as whole were performed to assess goodness of fit between models including and excluding variables for the primary outcome. 

Table 3, page 16, line 282-287. 

Univariable (model 1) and multivariable (model 2) logistic regression models of self-reported school-based mitigation measures (survey) and school-level probability of any positive COVID-19 case in school (SAIL). OR: Odds Ratio; 95% CI: 95% confidence intervals; p<0.05**, p<0.1*; adjusted for school size, proportion of pupils eligible for free school meals, clustered by school. Model 2 likelihood-ratio test keep two metres from pupils (p=0.1) and staff (p=0.03). See S3 Appendix for variable codebook. 

Finally, we have replaced 2-metre to two metre throughout the manuscript, and have updated table layouts.

---

## [Editor Report · Decision Letter 2]

2 Feb 2022

COVID-19 mitigation measures in primary schools and association with infection and school staff wellbeing: an observational survey linked with routine data in Wales, UK

PONE-D-21-27732R2

Dear Dr. Marchant,

We’re pleased to inform you that your manuscript has been judged scientifically suitable for publication and will be formally accepted for publication once it meets all outstanding technical requirements.

Kind regards,

Amitava Mukherjee, ME, Ph.D.

Academic Editor

PLOS ONE
---

## [Editor Report · Acceptance letter]

18 Feb 2022

PONE-D-21-27732R2 

COVID-19 mitigation measures in primary schools and association with infection and school staff wellbeing: an observational survey linked with routine data in Wales, UK 

Dear Dr. Marchant:

I'm pleased to inform you that your manuscript has been deemed suitable for publication in PLOS ONE. Congratulations! Your manuscript is now with our production department. 

Kind regards, 

on behalf of

Professor Dr. Amitava Mukherjee 

Academic Editor

PLOS ONE